# HtpG—A Major Virulence Factor and a Promising Vaccine Antigen against *Mycobacterium tuberculosis*

**DOI:** 10.3390/biom14040471

**Published:** 2024-04-11

**Authors:** Rita Berisio, Giovanni Barra, Valeria Napolitano, Mario Privitera, Maria Romano, Flavia Squeglia, Alessia Ruggiero

**Affiliations:** Institute of Biostructures and Bioimaging, IBB, CNR, Via Pietro Castellino 111, I-80131 Naples, Italy; giovanni.barra@unicampania.it (G.B.); valeria.napolitano@cnr.it (V.N.); mario.privitera@cnr.it (M.P.); maria.romano@cnr.it (M.R.); flavia.squeglia@cnr.it (F.S.)

**Keywords:** protein structure, tuberculosis, chaperone, folding, vaccine

## Abstract

Tuberculosis (TB) is the leading global cause of death f rom an infectious bacterial agent. Therefore, limiting its epidemic spread is a pressing global health priority. The chaperone-like protein HtpG of *M. tuberculosis* (Mtb) is a large dimeric and multi-domain protein with a key role in Mtb pathogenesis and promising antigenic properties. This dual role, likely associated with the ability of Heat Shock proteins to act both intra- and extra-cellularly, makes HtpG highly exploitable both for drug and vaccine development. This review aims to gather the latest updates in HtpG structure and biological function, with HtpG operating in conjunction with a large number of chaperone molecules of Mtb. Altogether, these molecules help Mtb recovery after exposure to host-like stress by assisting the whole path of protein folding rescue, from the solubilisation of aggregated proteins to their refolding. Also, we highlight the role of structural biology in the development of safer and more effective subunit antigens. The larger availability of structural information on Mtb antigens and a better understanding of the host immune response to TB infection will aid the acceleration of TB vaccine development.

## 1. Introduction

Tuberculosis (TB) is the world’s second-deadliest infectious disease after the coronavirus disease 2019 (COVID-19) pandemic, causing 1.3 million deaths and 10.6 million new cases in 2022 and is the leading killer among people with HIV. During the 2020–2022 period, COVID-19 disruptions resulted in nearly half a million more deaths from TB (https://www.who.int/publications/i/item/9789240083851, accessed on 15 March 2024). TB is caused by the ancient pathogen *Mycobacterium tuberculosis* (Mtb), which is transmitted through inhalation of aerosol particles and transported to the lungs, where it infects alveolar macrophages and can produce cavitation, a hallmark of TB infection due to collagen degradation resulting from strong immune response [1,2]. Infected cells are sequestered at the core of the TB granuloma, a confined niche where Mtb can persist in a dormant state for decades [3,4]. During dormancy, cells are not able to replicate and be detected in biological samples, but this non-culturable state is transient [5]. The reactivation from dormancy is the secret weapon of Mtb, as it can also occur after decades in about 10% of infected individuals. This complex phenomenon has been related to the catalytic action of resuscitation-promoting factors, a set of peptidoglycan-degrading proteins [6,7,8,9,10].

In activated macrophages and in other host microenvironments, Mtb can be exposed to multiple chemical stresses, including acidic pH, reactive oxygen, nitrogen species, antibiotics, and high temperature [11,12,13,14]. All of these conditions can be deleterious to cellular proteins, which can undergo unfolding and aggregation [15,16]. Despite this, Mtb is able to survive under stressful conditions for years [17].

In dormant cells, which can be reckoned as an extreme case of survival in stressful conditions, proteomic characterisation has shown significant differences in the number of enzymes and proteins which normally participate in the metabolic pathways [18,19,20,21]. Indeed, despite the negligible metabolic activity of dormant Mtb, several consensus proteins with a stabilising and protective effect, like chaperones DnaJ1 (Rv0352), HtpG (Rv2299), DnaK (Rv0350), GroEL2 (Rv0440), GroES (Rv3418), GroEL1 (Rv3417), have been found in different dormancy models [20,22]. Consistently, nonreplicating bacterial cells contain greater amounts of protein aggregates than those in an actively replicating state [16,23,24,25,26] due to the suboptimal function of protein quality control machinery, lack of nutrients to support protein synthesis, and an increase in cellular oxidants [27]. Therefore, a finely tuned chaperone expression level is crucial for Mtb’s stress response [14].

Due to its importance in Mtb virulence, the chaperone machinery is an attractive target for therapeutical applications against TB. Among chaperones, HtpG, which belongs to the highly conserved Hsp90 family proteins, was shown to induce strong activation of dendritic cells (DCs), enhancing the protective immune response when fused to other Mtb antigens (e.g., ESAT6) [28,29,30,31]. This protein acts as an actor in a complex plot, which requires the cooperation of several chaperone molecules [28]. A large number of structural studies of Mtb chaperones have been conducted in the last few years [30,32,33,34]. In this review, we provide an overall picture of the contribution of these structural studies to the understanding of the complex molecular machinery devoted to protein refolding and to the role of HtpG in this scenario.

## 2. HtpG of *M. tuberculosis*, a Mycobacterial Virulence Factor

HtpG of Mtb is a metal-dependent ATPase, fully conserved among Mtb strains and highly conserved in pathogenic mycobacteria like *M. leprae* (Table 1). Importantly, it is not encoded in avirulent species like *M. smegmatis*. HtpG was recently shown to be essential for maintaining the function of the CRISPR/Cas system of Mtb, which is involved in the defence against invading nucleic acids [35].

Although not essential, HtpG is important for maintaining proteostasis in stressed Mtb cells. Indeed, transposon mutagenesis has shown that the loss of neither the gene encoding for HtpG nor for ClpB (another Mtb chaperone) is lethal to Mtb. However, the depletion of genes encoding for both chaperones impairs Mtb recovery after exposure to host-like stress [14]. Sequence analyses show that HtpG shares 46% sequence identity with the Hsp90 chaperone of *Escherichia coli* (PDB code 2IOP), a finding that suggests a similar function for HtpG. Consistently, like Hsp90 of *E. coli*, HtpG was shown to bind both ADP and AMP-PNP with a µM binding affinity [31].

Although an experimental structure of HtpG is not yet available, computational studies have shown that HtpG adopts a highly stable dimeric structure, a property that is crucial for its functional role as a molecular chaperone [30]. HtpG is formed by three domains, which were shown to undergo large conformational variations from an open nucleotide-free to a compact ATP-bound state through a clamping mechanism (Figure 1).

The structural determinants governing these large conformational changes were suggested by Molecular Dynamics (MDs) approaches [30]. These studies have shown that ATP rigidly anchors an ATP-binding loop of the N-terminal domain, denoted as the “lid”. The ATP-bound lid was shown to exhibit a low level of flexibility due to the strong interactions mediated by the phosphate groups of ATP with lid residues (e.g., Gln125, Gly127) (Figure 2A). This conformation is the result of a conformational “switch” of the lid region from the structure of the free to the ATP-bound enzyme [30] (Figure 2A). Importantly, the conformational switch of the “lid” is not compatible with the structure of HtpG in its free state due to collisions of the “lid” with the middle domain (Figure 2B). Indeed, it was proposed that the low flexibility of the ATP-bound “lid” makes it unable to accommodate its structure compatibly with the orientation between middle and catalytic domains in the free enzyme [30].

The incompatibility of the ATP-bound conformation of the lid with the domain organisation of HtpG observed in the free state provided a structural rationale for the overall rearrangement of the enzyme in the presence of ATP (Figure 2C). The resulting clumping of HtpG due to ATP binding and the consequent lid switch is expected to allow HtpG to bind client proteins to facilitate their folding and then release them once folded through ATP binding [36,37,38,39].

## 3. HtpG Belongs to a Well-Orchestrated Chaperone Network

HtpG is part of a well-orchestrated and complex chaperone network, which includes the chaperone DnaK, its cofactors DnaJ1/J2, the release factor GrpE, and the protein disaggreases ClpB (Figure 3). In the complex scenario of protein rejuvenation after chemical stress, the disaggregase ClpB and the chaperone DnaK play an essential role in the early steps of protein recovery (Figure 3) [16,23,32,40].

ClpB is a large protein forming hexameric structures. ClpB proteins form a family of ATPases that disaggregate and solubilise aggregated proteins [41,42,43,44]. Similar to other prokaryotic ClpBs, ClpB of Mtb is composed of an N-terminal domain (NTD) and two nucleotide-binding domains (NBD1 and NBD2), which form a ring-shaped hexamer (Figure 3). DnaK is a highly conserved protein in prokaryotes. It is composed of an N-terminal nucleotide-binding domain (NBD) and a C-terminal substrate-binding domain (SBD) that are connected by a flexible linker [32,33]. It collaborates with the two cochaperones, DnaJ1 and DnaJ2 (Figure 3), which deliver non-native client proteins to DnaK and accelerate ATP hydrolysis by DnaK [45]. The NEF GrpE helps the release of tightly bound ADP through binding in 1:2 stoichiometry and promoting substrate release [33] (Figure 3). Importantly, an association of DnaK and ClpB at the aggregate surface stimulates ATP hydrolysis, which mediates substrate remodelling [16,23,40,46,47,48,49]. Consistently, Mtb cells lacking ClpB are sensitive to oxidants and show fault in recovery after achieving a stationary growth phase [14].

In many organisms, DnaK interacts with HtpG and ClpB [50,51,52], whereas in Mtb, DnaK interactions with ClpB [32] and with HtpG [14], but not directly between ClpB and HptG, have been reported. This suggests that ClpB and HtpG are part of a broader network of ATPases, all connected by DnaK. Intriguingly, while both HtpG and ClpB are not essential in Mtb [16,53], cells lacking both nonessential chaperones are hypersensitive to host-like stresses that induce a nonreplicating state [14]. This further suggests that HtpG acts cooperatively with the other chaperones (Figure 3).

**Figure 3 biomolecules-14-00471-f003:**
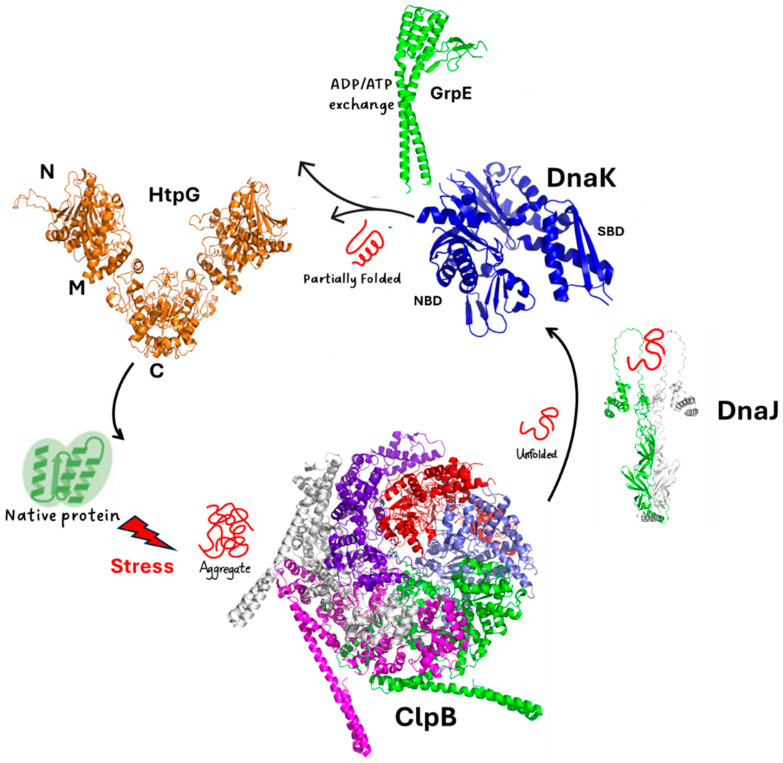
A scheme of the chaperone proteins assisting protein folding in Mtb. Cartoon representations are drawn for the experimental structures of ClpB (pdb code 7l6n) and DnaK (pdb code 8gb3). For the other proteins, we determined their structures using AI and the program AlphaFold2 [54], which provided a reliable model with pLDDT > 90.

## 4. A Structural Model of HtpG Interactions in the Chaperone Machinery of Mtb

Consistent with the observed interactions of HtpG with DnaK in Mtb [14], it was reported that an increased expression in the two cofactors of DnaK, DnaJ1, and DnaJ2, occurred in a ΔHtpG mutant strain [28]. Despite the importance of the HtpG–DnaK complex formation in Mtb, no structural models of this important interaction have been described so far. With the help of Artificial Intelligence (AI), we reliably model the HtpG–DnaK complex, as shown in Figure 4A,B. In addition to the typical parameters used to evaluate AI modelling reliability, we could validate this structure based on the available information from the homolog Hsp90 from *E. coli* [55]. Indeed, similar to Hsp90 of *E. coli*, HtpG interacts with DnaK exclusively through residues of its middle domain (Figure 4C), thus leaving the N-terminal domain free to exert its catalytic function. Importantly, we observe that the key residues involved in binding are strongly conserved between HtpG of Mtb and Hsp90 and between DnaK chaperones of the two bacteria, thus corroborating the similar organisation in Mtb (Figure 4D). As reported for Hsp90 [55], this structural organisation is mechanistically plausible, as it allows a client protein bound to the SBD of DnaK to readily interact with the client-binding site of HtpG (Figure 3). These studies provide insight into the interactions and collaboration between HtpG and DnaK.

A large amount of structural information on DnaK complexes was released in the last three years [32], among which the structure of DnaK with ClpB is illuminating [32]. Up to four DnaK molecules were found to bind to a ClpB hexamer, although the exact stoichiometry is unknown, and it is clear whether DnaK functions processively on ClpB or hops on and off [32]. We combined the structural information of the HtpG–DnaK complex with that of the DnaK–ClpB complex to generate a hypothetical model of the entire machinery (Figure 5). Although there is no experimental proof, it is tempting to propose that ClpB may bind up to six DnaK–HtpG complexes in a carousel-like machinery (Figure 5). In this machinery, which is so far only a putative model, the refolding process may proceed radially from the disaggregation of proteins operated by ClpB to refolding by DnaK and HtpG. Possibly, proteins that are partially refolded by DnaK are shuttled from the SBD domain of DnaK to HtpG on an adjacent arm of the carousel, closest in space. A similar mechanism would allow a high efficacy of the entire refolding machine.

## 5. HtpG as a Vaccine Antigen

As reported by the WHO, a new vaccine with better coverage than the current *Mycobacterium bovis* BCG vaccine is strongly needed [56]. There is extremely active research in the field, with a steep acceleration in the past decades due to the advance of technologies and more rational vaccine design strategies. Several vaccines against TB are in the pipeline, including live attenuated whole-cell vaccines, inactivated whole-cell vaccines, adjuvanted protein subunit vaccines, and viral-vectored vaccines [4]. Subunit vaccines contain purified parts of the pathogen that elicit immunogenic host response, and, as such, they overcome safety concerns typically associated with vaccines. Although they are specific and safe, they typically display a reduced capacity to stimulate a broad immune response compared to live attenuated or killed microorganisms. Therefore, these vaccines need to be engineered to enhance their immunogenicity. Structure-based antigen design is a rational approach that uses three-dimensional structural information to design novel and enhanced vaccine antigens. As previously mentioned, this strategy is crucial to the development of effective subunit antigens [57].

We have previously shown that HtpG is a promising vaccine antigen, as it activates dendritic cells (DCs), which can induce T-cell differentiation, resulting in inhibition of intracellular Mtb growth in macrophages [29,30,31]. The elucidation of the exact regions responsible for the immunoreactivity has helped the design of a more effective vaccine antigen [31]. Indeed, this step is an important tool for the design of improved antigens [57]. Crucial to the identification of epitope region is the prediction of the portions of the protein that are able to strongly bind the Major Histocompatibility Complex (MHC). The function of MHC molecules is indeed to bind peptide fragments derived from pathogens and display them on the cell surface for recognition by the appropriate T-cells. The human MHC class II is encoded by three different isotypes, HLA-DR, -DQ, and -DP (referred to as HLA in humans, which stands for human leukocyte antigens), all being highly polymorphic.

HtpG is predicted to embed many T-cell epitopes with a strong affinity to the MHCII complex, including HLA-DR, HLA-DP, and HLA-DQ molecules [31]. These epitopes are mainly located in the C-terminal domain of HtpG and some in the middle domain. No epitopes were predicted in the catalytic N-terminal domain. Consistently, we found that the most immune-reactive region of the molecule is located on the C-terminal and middle domains, whereas the catalytic and nucleotide-binding N-terminal domain plays no role in the elicitation of the immune response [31]. This information was precious to designing vaccine antigens with enhanced biophysical properties, albeit conserved or enhanced antigenic properties [31]. We also adopted a combination of different immune stimulation mechanisms upon fusion of middle and C-terminal domains of HtpG, which stimulates DC cells, with ESAT6, known as a T cell activator. This fusion antigen possessed a higher anti-mycobacterial activity than the fusion protein embedding the entire HtpG [31]. These studies point to HtpG as an important target for vaccine development and are in line with the dual function of Hsps in humans, depending on their intracellular or extracellular location. Intracellular Hsps have a protective function, whereas extracellular located Hsps mediate immunological functions [58,59].

## 6. Conclusions

TB has undergone times when it was considered a thing of the past. The development of the first antibiotics in the 1940s and 1950s led to an important decrease in TB case fatality. However, WHO’s Global Tuberculosis Programme (GTB) declared TB a global emergency in 1993 due to the development of antibiotic-resistant mycobacterial strains. The prevention of the development of drug-resistant strains of TB was its main goal 30 years later. This goal has not yet been achieved, and the identification of important targets for drug and vaccine development is a precious tool for the advancement of therapeutic approaches against TB.

The chaperone HtpG has a central role in Mtb virulence, as demonstrated by its conservation in virulent strains. In this review, we have presented the complex and dual role of HtpG in both protecting Mtb from stress and activating an immune response. We highlight the synergic role of HtpG with a large number of chaperones, which all together are responsible for the recovery of proteins after host-induced stress. Indeed, HtpG interacts with DnaK, which in turn interacts with the disaggregase ClpB. It is tempting to propose that these proteins form a large macromolecular machinery where ClpB exerts its role in the dissolution of protein aggregates and then shuttles disaggregated molecules to DnaK and HtpG for proper refolding (Figure 5). These steps, which occur through the help of cofactors like DnaJ1 and DnaJ2 and the release factor GrpE, involve extensive ATP hydrolysis, a process that allows the large conformational changes needed to fold and then release client proteins. Although this is a speculative model, it is based on the most updated structural findings [32,33]. We believe that the understanding of further structural aspects associated with protein folding by this complex machinery will be precious to the understanding of this key biological process, which is the ace in the hole of Mtb and allows it to survive in extremely stressful conditions. Also, we are increasingly aware of the fact that the understanding of the structural basis of immunogenicity is key to the design of improved antigens with better biophysical properties and the ability to induce multiple immune-stimulating mechanisms.

## Figures and Tables

**Figure 1 biomolecules-14-00471-f001:**
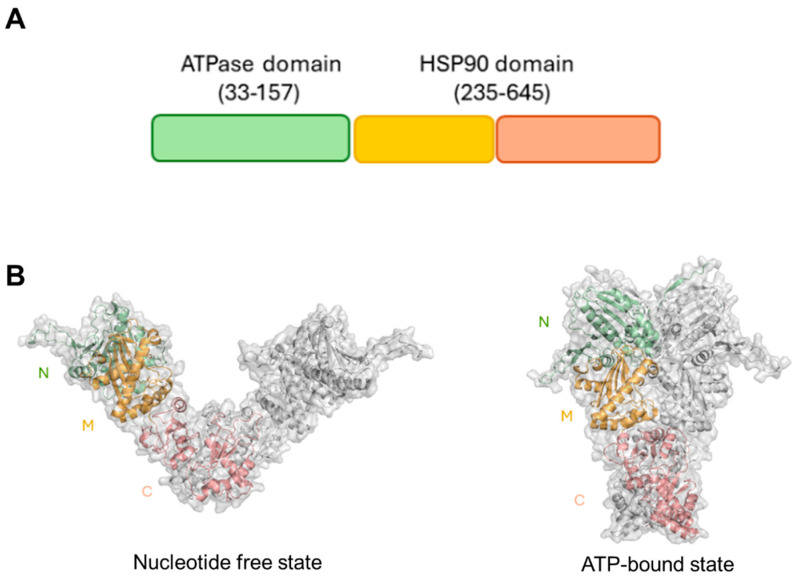
(**A**) Pfam organisation of HtpG. (**B**) Cartoon and surface representations of structural models of HtpG in its free state (on the left) and its ATPbound state (on the right). The two structures were derived by homology modelling, as previously reported [30]. Catalytic N–terminal, middle, and C–terminal domains are drawn in green, yellow, and salmon, respectively. Figures were generated with PyMol.

**Figure 2 biomolecules-14-00471-f002:**
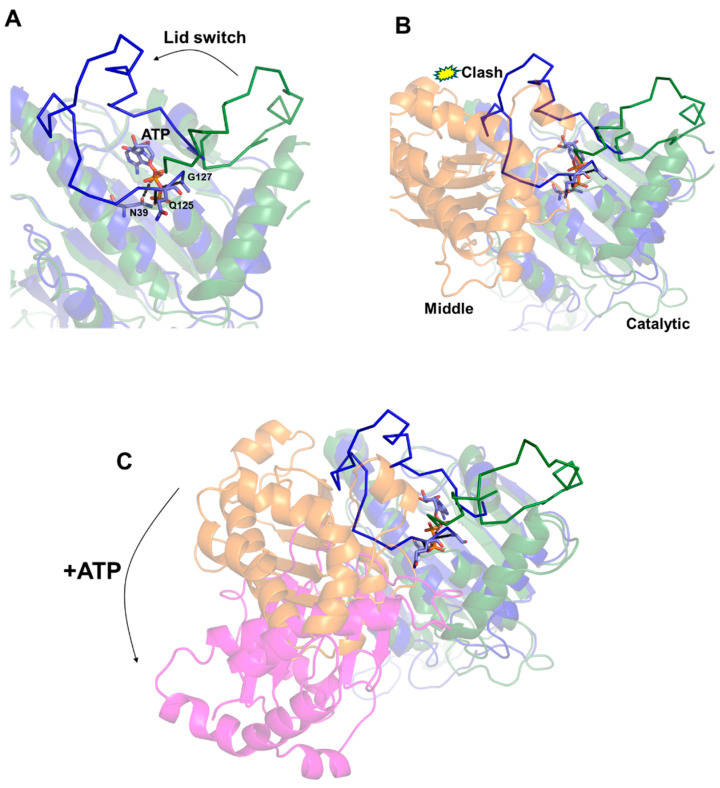
A cartoon representation of the structural modifications in the “lid” upon ATP binding in the computed structures of HtpG [30]. (**A**,**B**). The catalytic domains in the free and ATP-bound structures of HtpG are drawn in forest green and dark blue, respectively. The middle domain in the free enzyme in Panel B is drawn in orange. (**C**) Superposition of catalytic domains in the structures of free and ATP-bound HtpG [30], showing the 119° rotation of the middle domain upon ATP binding. The middle domains in the free and ATP-bound HtpG models are drawn in orange and magenta, respectively.

**Figure 4 biomolecules-14-00471-f004:**
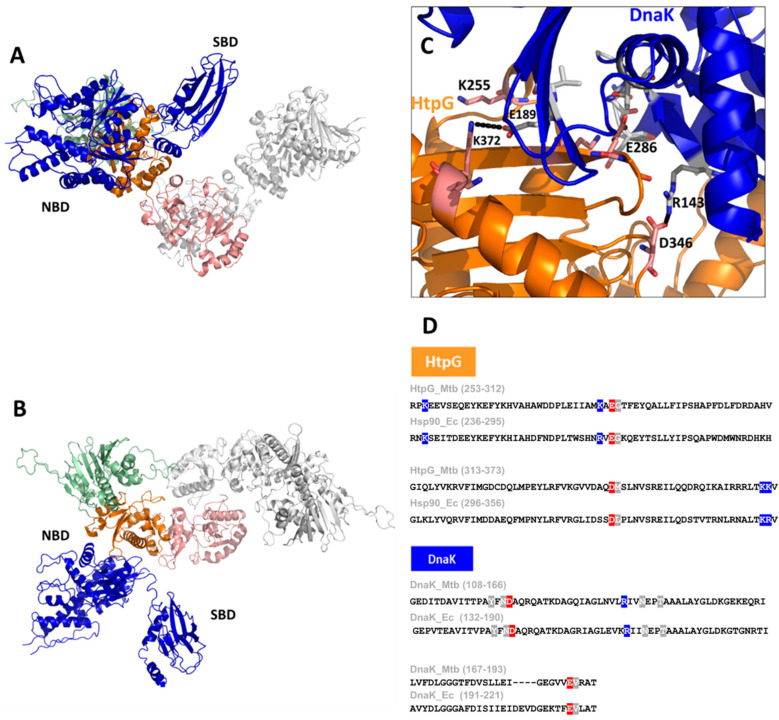
A cartoon representation of the AI-modelled structure of the HtpG–DnaK complex of Mtb (**A**,**B**). The structure was obtained using the software AlphaFold2.0 [54], which provided a reliable model with pLDDT > 90. A zoom of the interface between HtpG and DnaK is given in panel (**C**), with some important residues drawn in stick form. The sequence alignment of HtpG and DnaK of *E. coli* and Mtb are reported in panel (**D**), with the most important conserved residues highlighted (positives in blue, negative in red, uncharged in grey).

**Figure 5 biomolecules-14-00471-f005:**
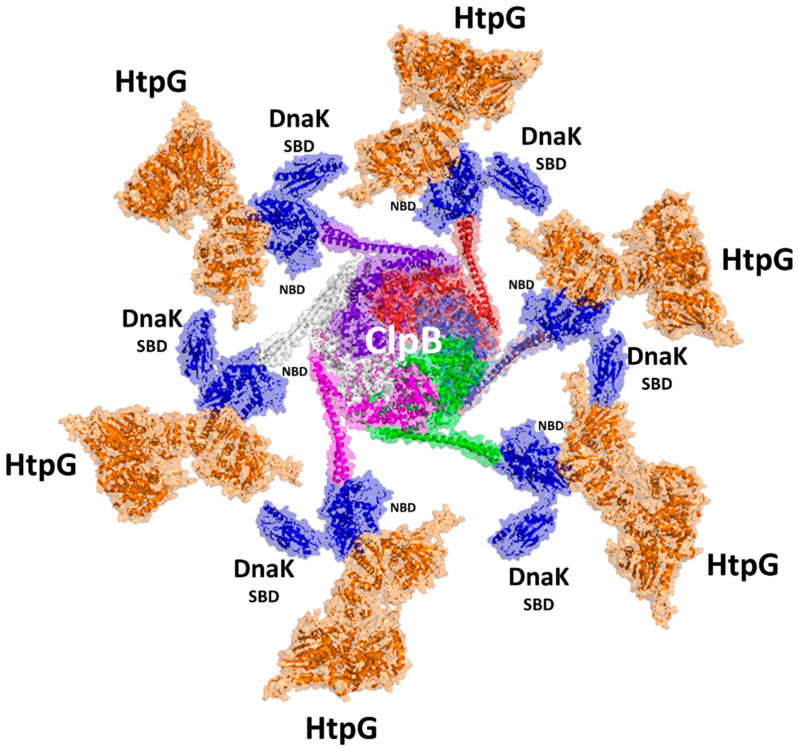
A model of the carousel-like chaperone machinery of Mtb. The model was generated upon superposition of the AI model of the HtpG–DnaK complex [54] to DnaK NBD in the cryoEM structure of the ClpB complex (pdb code 6w6e). Cartoon and surface representations are drawn for HtpG (orange), DnaK (blue), and the six chains of ClpB (white, prune, red, light blue, green, magenta).

**Table 1 biomolecules-14-00471-t001:** Sequence conservation of HtpG in mycobacterial strains.

Accession Code	Species	Query Cover	Sequence Identity (%)	N. Residues
WP_128886473.1	*Mycobacterium tuberculosis*	97%	99.84%	647
WP_036470126.1	*Mycobacterium triplex*	97%	86.08%	641
WP_065473763.1	*Mycobacterium malmoense*	97%	86.30%	644
WP_068157910.1	*Mycobacterium kubicae*	97%	85.60%	641
WP_085671006.1	*Mycobacterium szulgai*	97%	85.94%	646
WP_158017402.1	*Mycobacterium basiliense*	97%	87.26%	655
WP_069419255.1	*Mycobacterium intermedium*	97%	87.18%	645
WP_211695689.1	*Mycobacterium spongiae*	97%	84.91%	647
WP_071023730.1	*Mycobacterium talmoniae*	97%	85.02%	643
WP_191498542.1	*Mycobacterium simulans*	97%	87.11%	647
WP_085263439.1	*Mycolicibacter longobardus*	97%	83.60%	643
WP_144955623.1	*Mycobacterium helveticum*	97%	83.10%	650
WP_085076954.1	*Mycobacterium palustre*	97%	84.02%	642
WP_134429179.1	*Mycobacterium ulcerans*	97%	83.89%	644
WP_062538994.1	*Mycobacterium celatum*	97%	82.75%	642
WP_116540331.1	*Mycobacterium uberis*	97%	81.59%	656
WP_119607974.1	*Mycobacterium leprae*	97%	80.97%	656
WP_099038968.1	*Mycobacterium neglectum*	97%	81.42%	645
WP_043410932.1	*Mycobacterium rufum*	97%	79.97%	642
WP_083121049.1	*Mycolicibacterium rhodesiae*	97%	80.70%	641
WP_085108950.1	*Mycolicibacillus trivialis*	97%	80.38%	646
WP_163749624.1	*Mycolicibacterium helvum*	97%	79.43%	646
WP_240262914.1	*Mycobacterium paraterrae*	97%	76.81%	640
WP_211156151.1	*Mycolicibacterium septicum*	97%	77.19%	651
WP_085302105.1	*Mycobacterium koreense*	97%	77.71%	646
MBX7456299.1	*Mycolicibacterium* sp. 3033	97%	75.82%	646
WP_209922861.1	*Mycolicibacterium lutetiense*	97%	77.46%	650
WP_140691504.1	*Mycobacterium hodleri*	97%	75.08%	648
WP_085161654.1	*Mycobacterium lacus*	97%	86.57%	644
WP_075512658.1	*Mycobacterium ostraviense*	97%	88.21%	647
WP_113964043.1	*Mycobacterium shimoidei*	97%	83.54%	640
WP_065142642.1	*Mycobacterium asiaticum*	97%	83.96%	649
WP_085200206.1	*Mycobacterium fragae*	97%	83.57%	644
WP_085236188.1	*Mycobacterium conspicuum*	97%	82.28%	641
WP_163686705.1	*Mycolicibacterium gadium*	97%	81.89%	646
WP_163759712.1	*Mycobacterium botniense*	97%	81.55%	647
RAU95435.1	*Mycolicibacter senuensis*	97%	80.47%	649
WP_014208752.1	*Mycolicibacterium rhodesiae*	97%	80.44%	643
WP_097942061.1	*Mycolicibacterium agri*	97%	80.28%	644
WP_085195702.1	*Mycobacterium xenopi*	97%	80.06%	641
OLO99293.1	*Mycobacterium porcinum*	97%	78.16%	635

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
