# Peer review of "HtpG—A Major Virulence Factor and a Promising Vaccine Antigen against Mycobacterium tuberculosis"

_biomolecules, 2024, doi:10.3390/biom14040471_

Round 1
Reviewer 1 Report
Comments and Suggestions for Authors
The review by Berisio et al provides a comprehensive view of the structural determinants of the chaperone-like protein HtpG of Mycobacterium tuberculosis. The article describes present knowledge of the biochemistry and structure of this protein and provides relevant information about the mode of action of this protein in the large macro-complex with the complete chaperone machinery. Also, the potential role of this protein as a base for the design of novel vaccines against Mtb is described.
The manuscript is concise, the figures are clear, and the text is well-written. I have no relevant issues for this review. Just a minor comment on Figure 1 in which the origin of the structural models is not provided (while in the other cases, this point is described).
Author Response
Comment
The review by Berisio et al provides a comprehensive view of the structural determinants of the chaperone-like protein HtpG of Mycobacterium tuberculosis. The article describes present knowledge of the biochemistry and structure of this protein and provides relevant information about the mode of action of this protein in the large macro-complex with the complete chaperone machinery. Also, the potential role of this protein as a base for the design of novel vaccines against Mtb is described.
The manuscript is concise, the figures are clear, and the text is well-written. I have no relevant issues for this review. Just a minor comment on Figure 1 in which the origin of the structural models is not provided (while in the other cases, this point is described).
Response
We thank the reviewer for these positive comments. We have integrated the legend of the figure1, as suggested (red text).
Reviewer 2 Report
Comments and Suggestions for Authors
In Figure 1, the "B" should be positioned on the right (to mark the ATP-bound state).
The source of the structures needs to be mentioned in the Figure 1 legend.
Author Response
Comment
In Figure 1, the "B" should be positioned on the right (to mark the ATP-bound state). The source of the structures needs to be mentioned in the Figure 1 legend.
Response
We have corrected and integrated the legend of the Figure 1, as suggested (red text). As for the position of “B”, this is intended for both homology models (free and ATP-bound) whereas “A” refers to PFAM domain organisation, as described in the legend. This is the reason why we have positioned it on the left.
Reviewer 3 Report
Comments and Suggestions for Authors
Comments and Suggestions:
The review article " HtpG, a Major Virulence Factor and a promising vaccine antigen against Mycobacterium tuberculosis" by Berisio et al., is well-written and presented clearly. This review discusses the latest findings on HtpG structure and function, emphasizing its collaboration with Mtb chaperone molecules to assist in Mtb's recovery from host-like stress by facilitating protein folding. Additionally, it underscores the significance of structural biology in developing improved subunit antigens for TB vaccines, which will be enhanced by increased availability of structural information on Mtb antigens and a deeper understanding of the host immune response to TB infection.
1. Table 1: The table can be modified in a presentable form with upper lines and font size should match with the text in the manuscript. Species name should be italicized.
2. Figure 1: the fonts size of the A, B, C should be uniform in whole manuscript.
3. Figure 3, line 148: How reliable is the modeled figure generated with AI? Which tool was used. Please add any reference that generated the AI modeled figures.
4. Figure 3, line 148 and Figure 4, line 169: ‘AlphaFold’ and ‘AlphaFold2.0’, are same versions of the software was used to generate the figures?
5. Figure 4, panel D: The font size can be made bigger.
Author Response
Comment
The review article " HtpG, a Major Virulence Factor and a promising vaccine antigen against Mycobacterium tuberculosis" by Berisio et al., is well-written and presented clearly. This review discusses the latest findings on HtpG structure and function, emphasizing its collaboration with Mtb chaperone molecules to assist in Mtb's recovery from host-like stress by facilitating protein folding. Additionally, it underscores the significance of structural biology in developing improved subunit antigens for TB vaccines, which will be enhanced by increased availability of structural information on Mtb antigens and a deeper understanding of the host immune response to TB infection.
- Table 1: The table can be modified in a presentable form with upper lines and font size should match with the text in the manuscript. Species name should be italicized.
Response
We thank the reviewer for these positive comments. As suggested, we modified Table 1 to uniform it to the rest of the text
Comment
- Figure 1: the fonts size of the A, B, C should be uniform in whole manuscript.
Response
DONE
Comment
- Figure 3, line 148: How reliable is the modeled figure generated with AI? Which tool was used. Please add any reference that generated the AI modeled figures.
Response
Comment
- Figure 3, line 148 and Figure 4, line 169: ‘AlphaFold’ and ‘AlphaFold2.0’, are same versions of the software was used to generate the figures?
Response
We used AlphaFold2 in all cases to calculate protein structures. These computational tools are both based on the modern Deep Learning technique concept, with AlphaFold2 showing a higher accuracy.
Comment
- Figure 4, panel D: The font size can be made bigger.
Response
We have replaced this figure.
Reviewer 4 Report
Comments and Suggestions for Authors
General comment
Tuberculosis (TB), its pathogenesis, and the role of the chaperone-like protein HtpG in Mycobacterium tuberculosis (Mtb) survival and virulence.
some comments for improvement:
Clarity and Structure: The introduction effectively sets the stage by presenting key statistics about TB and highlighting its significance compared to other pathogens.
References: It's commendable update references.
Some sentences could be condensed for clarity and brevity without losing essential information. For instance, "During dormancy, cells are not able to replicate and to be detected in biological samples, however, this non-culturable state is transient" could be simplified to "During dormancy, Mtb cells are non-replicating and undetectable in biological samples, but this state is transient."
Ensure consistent terminology throughout the introduction. For instance, you refer to "Chaperones" and "chaperone molecules" interchangeably. Maintaining consistency in terminology enhances clarity for the reader.
Figure and Table References: When referring to figures and tables within the text, ensure to include the corresponding figure or table number.
Language and Grammar: Review the text for any grammatical errors and ensure that the language used is clear and concise.
Comments on the Quality of English Language
Language and Grammar: Review the text for any grammatical errors and ensure that the language used is clear and concise.
Author Response
Tuberculosis (TB), its pathogenesis, and the role of the chaperone-like protein HtpG in Mycobacterium tuberculosis (Mtb) survival and virulence.
some comments for improvement:
Clarity and Structure: The introduction effectively sets the stage by presenting key statistics about TB and highlighting its significance compared to other pathogens.
Some sentences could be condensed for clarity and brevity without losing essential information. For instance, "During dormancy, cells are not able to replicate and to be detected in biological samples, however, this non-culturable state is transient" could be simplified to "During dormancy, Mtb cells are non-replicating and undetectable in biological samples, but this state is transient."
Ensure consistent terminology throughout the introduction. For instance, you refer to "Chaperones" and "chaperone molecules" interchangeably. Maintaining consistency in terminology enhances clarity for the reader.
Figure and Table References: When referring to figures and tables within the text, ensure to include the corresponding figure or table number.
Language and Grammar: Review the text for any grammatical errors and ensure that the language used is clear and concise.
Response
We thank the referee for this supportive comment. We have shortened some sentences and proof-read the text. We also improved most of the figures.